# A Deep Learning Method of Human Identification from Radar Signal for Daily Sleep Health Monitoring

**DOI:** 10.3390/bioengineering11010002

**Published:** 2023-12-20

**Authors:** Ken Chen, Yulong Duan, Yi Huang, Wei Hu, Yaoqin Xie

**Affiliations:** 1Shenzhen Institute of Advanced Technology, Chinese Academy of Sciences, Shenzhen 518055, China; k.chen@siat.ac.cn (K.C.); yl.duan@siat.ac.cn (Y.D.); 2Shenzhen HUAYI Medical Technologies Co., Ltd., Shenzhen 518055, China; huangyi@huayimt.com (Y.H.); huwei@huayimt.com (W.H.)

**Keywords:** human identification, millimeter wave radar, deep learning

## Abstract

Radar signal has been shown as a promising source for human identification. In daily home sleep-monitoring scenarios, large-scale motion features may not always be practical, and the heart motion or respiration data may not be as ideal as they are in a controlled laboratory setting. Human identification from radar sequences is still a challenging task. Furthermore, there is a need to address the open-set recognition problem for radar sequences, which has not been sufficiently studied. In this paper, we propose a deep learning-based approach for human identification using radar sequences captured during sleep in a daily home-monitoring setup. To enhance robustness, we preprocess the sequences to mitigate environmental interference before employing a deep convolution neural network for human identification. We introduce a Principal Component Space feature representation to detect unknown sequences. Our method is rigorously evaluated using both a public data set and a set of experimentally acquired radar sequences. We report a labeling accuracy of 98.2% and 96.8% on average for the two data sets, respectively, which outperforms the state-of-the-art techniques. Our method excels at accurately distinguishing unknown sequences from labeled ones, with nearly 100% detection of unknown samples and minimal misclassification of labeled samples as unknown.

## 1. Introduction

Sleep is a crucial biological process. Transient sleep pattern changes and disorders, such as acute sleep deprivation, sleep breathing disorders, narcolepsy, periodic limb movement syndrome, REM sleep disorder, and parasomnias, have been reported to have a significant impact on people’s health, sometimes even leading to lethal consequences [1,2]. Consequently, monitoring sleep is considered as a task with clinical significance and practical prospects.

The development of sensor technologies has made it possible to extend the range of sleep monitoring from professional clinical environment to daily home-monitoring scenarios. To establish long term health monitoring and conduct data analysis, human identification of the acquired signals plays a crucial and prerequisite role, and has recently become a prominent requirement along with the growing application of intelligent healthcare devices in daily life. Inaccurate identification may cause failure in health monitoring and disease warning, and affects the trustworthiness of the technologies to be applied in the real world. 

Vision-based solutions such as videos and pictures have been widely used for human identification. Although effective, such surveillance systems have their limitations. They are sensitive to environment light condition changes, and are intrusive methods where privacy issues could be aroused [3,4]. 

Various studies have chosen radar signals as a non-contact source for vital biometric extraction, motion monitoring, and pose estimation [5,6,7,8,9,10,11], which suggests that, from radar data, we can acquire highly personal information, which is potentially promising for human identification. Some groups [12,13,14] have shown that with machine learning techniques, the features of motions such as walking and running can be promising for trajectory identification and tracking. The radar signals of different participants showed a distinguishable feature difference in range-Doppler (RD)/range-Doppler-azimuth (RDA) map and micro-Doppler (μD) signature, which can be recognized by machine learning techniques. However, such features may not be feasible during stationary scenarios such as resting and sleeping. To stress such limitations, some groups used radar to acquire heart motion [15,16] or respiratory pattern [17,18] for human identification. However, there were certain constraints on the data acquisition setups. The data sets were usually acquired in the seated position. Or, laser positioning was introduced to ensure the precise position of the antenna. Such ideal acquisition conditions may not be achievable during actual home-monitoring scenarios. When the acquired heart motion signal sequences are mixed with noise and random body movements, the feature signatures in Short Time Fourier Transform (STFT) images or the single beat segments may not be as obvious as described in the literature. In this work, we will explore a solution of a time series classification problem for radar signal sequences, and discuss the factors which may affect the accuracy of the classification in details.

When discussing the human identification problem from radar data, most of the studies tend to evaluate the performance of the proposed methods using the close set classification accuracy with labeled data. However, in practice, the radar signal sequences acquired from actual scenarios are quite different from the data for training. Such unknown sequences cannot be recognized by the model and therefore not correctly labeled, causing confusion and a potential data safety hazard. To tackle this issue, a more robust scheme should be proposed to detect and identify such data properly. 

One intuitive solution is to assign the unknowns an additional label, turning the unknown detection problem into a classification problem so that supervised learning methods can be applied. Such methodologies have their limits. The features of the unknown samples cannot be exhaustively numerated; therefore, open set recognition methods are studied. Traditional machine learning methods [19,20,21,22,23] such as SVM, sparse representation, and nearest neighbor have been proposed to be utilized in this area. However, such methods suffered from the common limitation, and the necessity of carefully selected features. Deep learning-based methods have shown powerful potential in self-adaptive feature extraction and recognition. Open Max [24,25] was proposed to modify the softmax layer in the deep learning model, which used Weibull-based calibration to weight the classification scores and add a new score for the unknown class. Generative and adversarial methods [26,27,28], auto-encoder and its variants [29,30] were also reported to model the features of the unlabeled data. Besides such general methods, there were also schemes specially designed for open set recognition [31,32]. These methods usually required specifically designed network structures or learning schemes, and were primarily focused on nature images. As far as we know, until now, the open set recognition problem, especially on radar signal data, is still a challenging task which has not been sufficiently studied. 

In this paper, we propose a deep learning-based scheme for open set human identification from radar signals, and consider human identification as a classification problem. The radar signal sequences are preprocessed and divided into fixed length samples for training. We represent each radar signal sample as a point in the proposed feature space, so that the samples from unknown sequences can be distinguished from those from the labeled sequences.

The rest of this paper is organized as follows. In Section 2, we explain the details of the proposed identification scheme. In Section 3, we discuss the performance of the proposed method by comparing with state of the art methods. In Section 4, we discuss the implementation in detail and the robustness performance. And the conclusion is in Section 5.

## 2. Materials and Methods

### 2.1. Identification Scheme

In this section, we will describe our proposed human identification scheme in detail. The overall structure of the scheme is briefly illustrated in Figure 1.

The training process is illustrated with the blue lines. The radar sequences are processed and divided into sample sets which will be described in detail in Section 2.2, and fed into deep convolution network for close set classification. Along with the probability distribution vector, we also extract the input to the last Dense N layer as interested feature vectors, and generate a Principal Component Space (PCS). Each radar sequence will be represented as a point cloud in this space. When a new unlabeled sequence is acquired, the labeling process follows the red lines. The sequence is preprocessed and fed into the trained network to extract the feature vectors. The feature vectors are then projected into PCS to generate the point cloud representation. We compare the features of the point cloud of the unlabeled sequence with those of the labeled sequences, and decide whether it is an unknown sequence. If the features of the unlabeled sequence agrees with one of the training sequences, the Dense N and softmax layer gives out the probability distribution and the corresponding label is assigned as a normal classification network. Otherwise, one category of the training sequences is replaced with the unknown sequence, then the network is retrained and PCS is updated. Repeating such cycles with acquired sequences, the network can be gradually customized by the user data.

### 2.2. Data Processing and Data Set Composition

A typical radar signal sequence is shown in Figure 2a, which is a time sequence showing the obstacle movement relative to the radar device. Each sequence records the movement of the human body continuously in a time period about several tens of minutes. An id is assigned to each sequence to distinguish different person in the acquisition procedure.

We can see that the sequence contains static baseline shift, representing the distance between person and the radar, and linear slopes which suggest position change process of human body relative to the device during data acquisition. It also contains small periodical fluctuations representing the unconscious movement of the human body, such as respiration and heartbeat. We introduce a PELT [33]-based change point detection method to divide the sequence into segments according to the change points of slopes, and apply a linear fitting for each segment which is then subtracted from the original sequence to eliminate the large linear components. A typical result is shown in Figure 2a, and detailed comparisons of sequences before and after the baseline shift and slope suppression are provided in Figure 2b,c. The impact of changes in the data acquisition environment is suppressed, and only the relatively weak motion is left, which represents the signature of the detected person.

After the preprocessing step, the sequence shows approximately periodical fluctuations without baseline shift and large scale slopes, representing a learnable mixed pattern of respiration and heartbeat. To form the data sets, we divide the preprocessed sequences into fixed-length samples, and assign each sample a label by the id of the corresponding sequence. The length of the samples should be properly chosen. A too small sample length may not cover a complete period of the sequences, undermining the integrity of the signal features, while a too long sample length will require more GPU memory, and may affect real time performance. A trade off should be made between accuracy and efficiency, and the choice of sample length will be discussed in Section 4.1.

### 2.3. Deep Learning Model Summary

The details of the convolution neural network in Figure 1 is shown in Figure 3. The network is composed of 3 sequential convolution blocks, each containing 2 repeated one-dimensional convolution layers followed by a ReLu activation layer. Between convolution blocks, max pooling layers are used for down sampling. As the depth of the blocks increases, we gradually increase the number of convolution layer channels in the blocks, and decrease the size of the convolution kernel. The numbers of filters of the repeated convolution layer in each block are 32, 64, and 128, respectively. The kernel sizes in each block are 7, 5, and 3, respectively. The number of strides is 1. The kernel size of the max pooling layer is 3. A dense 128 layer is used to generate the classification probability distribution from the features extracted by convolution blocks. 

### 2.4. Unknown Detection and Principal Component Representation

We utilize the features extracted by the close set classification network as a criterion for unknown sample detection. 

After the close set classification training is completed, the samples $\left\{s_{i,j}\right\}$ in the training data sets are fed into the network. The input tensors to the last dense layer are extracted to form the key feature sets $\left\{f_{i,j}\in R_{1\times d}\right\}$ where i=1,2,…,C, $j=1,2,\ldots ,N_{i}$, and C is the number of the persons in the training data sets. $N_{i}$ is the size of the training data set with label i, d is the size of the features. 

We apply PCA to the key feature sets, and obtain the principal components $\left\{v_{k}\in R_{d\times 1}\right\}$ where k=1,2,…,d, which forms the basis of a space describing the key feature sets, which we define as PCS. An arbitrary feature $f_{i,j}$ of the sample $s_{i,j}$ in the training data sets can be represented as a d-dimension point in PCS $P_{i,j}=[p_{i,j,1},p_{i,j,2},\ldots ,p_{i,j,d}]$, where
(1)$$p_{i,j,k}=f_{i,j}\cdot v_{k}$$

We cluster the points by their respective label, and calculate the mean $\mu_{i}$ and standard variance $std_{i}$, which describe the statistics index of the labeled data sets in PCS representation. For an unlabeled sequence, we follow the same procedure, and feed an arbitrary sample $s_{j_{unlabeled}}$ to the network to generate the feature $f_{j_{unlabeled}}$. We describe the feature as a point in PCS as $P_{j_{unlabeled}}$ according to Equation (1). If $P_{j_{unlabeled}}$ does not follow the PCS statistics index of any labeled data sets, we mark the sample as an unknown sample. Otherwise, the dense N and softmax layer continues to give out the probability distribution and one-hot vector as the normal classification network does.

The advantage of the proposed unknown detection method is that it makes full use of the byproduct of the classification network. The unknown sample detection and normal sample labeling are completed simultaneously, requiring no additional specifically designed network structures or extra learning schemes to capture the feature difference between the unknown and the labeled data.

## 3. Experiment and Results

### 3.1. Experiment Setup and Implementation Details

The data we used to test the proposed method in this paper come from two sources.

The first source is the public Schellenberg data set [34] of clinically recorded data which contains vital signs measured through radar and the synchronized reference sensor. The radar system used in this data set is based on Six-Port technology and operates at 24 GHz in the Industrial Scientific Medical (ISM) frequency band. It monitored 30 healthy volunteers in the following scenarios: Resting, Valsalva Maneuver (VM), Apnea, and Tilt Table Up and Down. For each volunteer, each scenario was measured for about 10 min. A positioning laser is used to align the antenna with the interested region. Meanwhile, a Task Force Monitor (TFM) is used as the reference device to measure the electrocardiogram, impedance cardiogram, and non-invasive continuous blood pressure. An image cited from the literature [31] shows the data acquisition setup in Figure 4a. In this study, we are primarily interested in the human identification in stationary states; therefore, we randomly selected 10 sequences of the cardiac radar data in the resting scenario from this public data set for subsequent experiments.

The second source is an experimentally acquired data set in our lab with a millimeter wave radar operated in Frequency Modulated Continuous Wave (FMCW) mode. The device working parameters are listed in Table 1. The radar device is placed on the nightstand table, with a tilted downward position, which ensured the torso is in the boresight of the radar. Currently, there is no specific orientation and position requirements of the device. We will investigate the influence of device position and orientation in our future work. We monitored the radar signal data of four volunteers in a sleeping state. The volunteers are asked to remain in a stationary supine posture, for about 40 min. In this experiment, no laser positioning is used. Informed consent was obtained from all the volunteers. The experimental procedure for data acquisition and utilization in this research has been approved by the ethics committee of Shenzhen Institute of Advanced Technology, Chinese Academy of Sciences. The committee reference number is SIAT-IRB-231011-H0675. This data set simulates a daily home-monitoring scenario, where the ideal data acquisition setup of the Schellenberg data set is not applicable. Figure 4b shows a typical data acquisition environment setup.

The proposed deep learning method is implemented with PyTorch on an NVIDIA GeForce RTX 3060 GPU. We choose the Adam optimizer to minimize the cross entropy. We introduce a learning rate decay schedule; the learning rate is reduced by 0.5 times when the metric stops improving.

### 3.2. Identification Accuracy

To test the performance of the proposed human identification network, we randomly select 20% samples from each sequence to form a test subset, and use the remaining samples to train the deep learning model with a validation split ratio of 0.1. Then, we have the training, validation, testing distribution to be 72:8:20. For the public data set, we randomly choose the data of 10 volunteers. When we choose the sample length of 2 s, the number of total samples of each volunteer is 300, as a typical sequence is about 10 min long. For experimentally acquired data sets, the data of four volunteers are used. Similarly, the number of total samples of each volunteer is set to 1000. We evaluate the classification accuracy by the confusion matrices, and compare the performance of our proposed network with state-of-the-art deep learning techniques.

We show the identification accuracy of our proposed method by the typical confusion matrices shown in Figure 5. For the public data set and the experimentally acquired data set, the average labeling accuracy reaches 98.2% and 96.8%, respectively.

We also compare the proposed method with other machine learning techniques, such as the LSTM-based classification network and Inception Network specially designed for time sequence classification tasks as proposed in [35]. The Inception Network applies multiple filters with varying length simultaneously to the input time series, which can automatically capture the features from both long and short time series. For the public data set, the LSTM classification network achieves 97.6% labeling accuracy, and the Inception Network achieves 97.5% labeling accuracy, respectively. For the experimentally acquired data set, the LSTM achieves 90.2% labeling accuracy and the Inception Network achieves 96.5% labeling accuracy, respectively. We also show typical result confusion matrices in Figure 5.

For the public data set, the labeling accuracy of the three methods is of the same level. For the experimentally acquired data set, since the data acquisition conditions are not as ideal as the public data set, the labeling accuracy of the LSTM classification network falls significantly, while the labeling accuracy of the two convolution-based methods (our proposed method and Inception Network) is affected less. Therefore, we conclude that the labeling accuracy of our proposed method outperforms the LSTM-based classification method, and achieves an equivalent level of accuracy with the Inception Network. However, the parameter size of our proposed method is obviously smaller than the Inception Network with less calculation time, as shown in Figure 6. We can also see that although, when dealing with more complex data with less obvious periodical patterns, the performance of LSTM classification network is lower as compared with convolution-based methods, the parameter size is also significantly smaller with a higher calculation efficiency, which suggests potential advantages in scenarios with high real-time requirements.

### 3.3. Unknown Sequence Detection Accuracy

To test the detection of unknown samples, we randomly select one sequence as the unknown sequence, and use the remaining sequences as labeled sequences to generate the labeled data sets to train the network and generate the PCS representation. The training data set, validation data set and the test subset are split with the same ratio (72:8:20) as the close set human identification problem mentioned in Section 3.2. The samples of the unknown sequences and the test subset are mixed to form the unknown test sample set, so that we can evaluate how our proposed unknown detection method works while preserving the feature recognition of the labeled sequences. When the training process completed, we feed the unknown test sample set to the network, express the samples as points in the PCS representation, and assign the “unknown” or “labeled” label according to the criterion described in Section 2.4.

To examine how the unknown samples can be distinguished from the labeled data, under the principal component representation, we define four statuses to describe the unknown detection result. If an arbitrary sample from the unknown sequence is detected as ‘unknown’, it is defined as True Positive (*TP*); otherwise, it is defined as False Negative (*FN*). If an arbitrary sample from the labeled sequences is detected as ‘labeled’, it is defined as True Negative (*TN*); otherwise, it is defined as False Positive (*FP*). The metrics to evaluate the unknown detection accuracy are given as follows:(2)$$accuracy=\frac{TP+TN}{TP+TN+FP+FN}$$
(3)$$precision=\frac{TP}{TP+FP}$$
(4)$$recall=\frac{TP}{TP+FN}$$
(5)$$FPR=\frac{FP}{TN+FP}$$
(6)$$F1=\frac{2\times precision\times recall}{precision+recall}$$

The accuracy shows the general labeling accuracy of the entire unknown test data set. The precision evaluates, among all the samples detected as “unknown”, how many of them are actually from the unknown sequence, and how many of them are falsely marked “unknown” ones. A higher precision shows better ability to distinguish the feature difference between unknown and labeled sequences. The recall evaluates how many of the actual unknown samples are accurately detected. A higher recall shows better ability to detect the unknown sequences. The FPR evaluates how many of the labeled samples are inaccurately marked as unknown samples. A smaller FPR shows better ability to recognize the labeled sequences. The F1 score is used to balance the precision and recall when the class distribution is unbalanced.

For visualization purposes, we use the first three principal components to show the feature distribution of the labeled and unknown samples in Figure 7. It can be seen that although only 85% of the total variance is utilized, the distribution of the point clouds still shows an apparent difference in the PCS.

The accuracy of the unknown data detection is shown in Table 2. We report the recall to be about 100%, which suggests almost all the unknown samples are correctly marked. A very small FPR and high precision suggests that there is very little part of the labeled samples inaccurately marked as unknown. Since the sample set is generated from a continuous time sequence, occasional sample detection error can be filtered out in time domain; therefore, it will not affect the final decision of unknown data detection. In the future, we will explore a detailed motion-tracking and time domain-filtering scheme to further improve the robustness of unknown detection scheme.

We compare our method with the widely applied open set recognition method—OpenMax. OpenMax is based on the Extreme Value Theory to incorporate likelihood of the recognition system failure, by fitting a Weibull distribution to the samples of each known class, and to correct the known class score and calculate an unknown class score by the probability of classification failure. When the maximum classification probability is obtained in the unknown class or the maximum classification probability is less than a certain threshold, the unknown class is recognized. As a widely applied open set recognition method, OpenMax is an extension of the existing classification network, and does not require extra specifically designed network structures or additional learning schemes.

We report that the detection rate of unknown samples by OpenMax is around 90%, but we also notice that this method affects the accuracy of close-set human identification. The average identification accuracy falls to around 87.0%.

The correction procedure of OpenMax changes the information extracted from the deep learning model; therefore, the identification accuracy might be affected. OpenMax does not improve the feature representation, and the fitting criterion is a handcrafted function which is not involved in the training; therefore, it might not always fit the right distribution of the feature space. Therefore, it may be more difficult to distinguish the unknown samples from the labeled ones when the features are more complicated. Meanwhile, our proposed method utilizes the features extracted by the network to generate the Principal Component Space. When the features are mapped into the generated space, the difference of distributions is improved, while the information extracted by the network is maintained.

Therefore, we conclude that although OpenMax has been proved promising for natural images, when applied to a time series such as the radar sequences, the performance may not be as accurate and robust. Figure 8 shows a typical confusion matrix comparison.

## 4. Discussion

### 4.1. Parameter Selection

We look into the choice of hyper parameters: the length of the sample as described in Section 2.2. We tend to choose a smaller length to improve the real-time performance, which might be critical for fatal disease detection and warning, and also release the calculation burden of the network. Figure 9 shows the performance changing against the sample length. We can see that when the sample length is too small (1 s), the labeling accuracy is remarkably lower. The reason is that a 1 s sample may not completely cover a typical radar motion cycle, causing certain time domain feature corruption. When the sample length is long enough (>2 s), the labeling accuracy does not change significantly, but the calculation time increases as the sample length increases. The heart rate is commonly 40–120 beats per minute; therefore, we experimentally choose the optimal sample length as 2 s.

### 4.2. Robustness Analysis

We show the necessity of the preprocessing step to suppress the impact of changes in the data acquisition environment in the radar signal as stated in Section 2.2. To simulate different radar positions, we add different offsets to the original sequences. And to simulate the unexpected position change process between the human body and the device, we add a linear component of random slope to each segment of the original sequences. The boundary of the random slope is set to be within a certain ratio of the original slope. The id’s of the test sequences remain unchanged since, when the same person is placed under different scenarios, the changes of the data acquiring environment should not affect the personal identification features. We train the model with the original sequences, and label the generated test sequences, both without the preprocessing step.

Figure 10a shows typical examples of the two categories of the test sequences. We gradually increase the offset, and the ratio of the random slopes, respectively, and summarize the labeling accuracy in Figure 10b,c. We can see that as the value of offset increases, the accuracy of labeling gradually decreases. We also observe that even a very slight change in the slope may affect the labeling accuracy significantly. This means that when the linear components are not removed from the sequences, the spontaneous and cyclic motion, which is more likely to represent the physiological features, is neglected due to its relatively low amplitude, while the interactions between the human body and the data acquisition environment affect the network training significantly and cause an inaccurate result when the scenario changes.

Therefore, the proposed preprocessing step is necessary to ensure the robustness of the human identification network. Without the preprocessing step, the network will be affected by the features of the environment changes due to its large amplitude, rather than focusing on the minor but representative personal signatures, causing the inaccurate identification result.

### 4.3. Impact of Number of People

We first study how the number of people affects the identification accuracy. The network is tested with randomly chosen 4, 6, 8 and 10 sequences. The identification accuracy against the number of people is shown in Table 3. We can see that when the number of people differs, the identification accuracy performance does not differ significantly. The fluctuation of the number of the correctly identified samples is only one, and the maximum number of inaccurately identified sample is less than two on average.

We also show how the number of people in the labeled data sets affects the performance of the unknown sequence detection methods. We evaluate three key performance metrics. The Labeled Sample Identification Accuracy demonstrates how the open set condition affects the close set identification accuracy. The Unknown Sample Detection Accuracy shows how many of the unknown samples can be accurately detected. The Unknown Sample Detection FPR shows how many labeled samples are inaccurately detected as unknown. We gradually increase the number of people from 4 to 10, with an increase of 2 people per step. We show the comparison of the performance metrics of the proposed method and the state-of-the-art methods against the number of people in Figure 11.

As the number of the people increases, it is more difficult for the OpenMax to fit the correction weights to be applied to the network inferred probability distribution, therefore causing labeled samples to be inaccurately detected as unknown, and also causing more identification errors within the labeled data sets. Therefore we can observe from Figure 11 that the performance of OpenMax fluctuates, while the performance of the proposed PCS method remain robust. The proposed method also achieves higher accuracy of the labeled sample identification and the unknown sample detection, and fewer labeled samples are inaccurately detected as unknown samples. It is demonstrated that with the proposed PCS representation, the unknowns can be accurately distinguished from the labeled data, while the identification accuracy of the labeled data is preserved.

## 5. Conclusions

In this paper, we propose a novel deep learning-based scheme for human identification using radar sequences. In addition to employing traditional close-set classification methods, we extract input tensors to the final dense-N layer as a feature of interest, introducing a Principal Component Space presentation. In the Principal Component Space, the radar sequence features can be represented as point clouds, and the feature difference among labeled sequences and the unknown sequence can be distinguished.

We use the public Schellenberg data set and a data set experimentally acquired in our lab to validate our proposed method. We compared our proposed method with state-of-the-art deep learning methods. Compared with LSTM network, our proposed approach demonstrates superior accuracy and robustness. Compared with the Inception Network, another convolution based state-of-the-art method, our proposed approach achieved an equivalent level of accuracy with less computation time and fewer parameters. This suggests that our proposed approach outperforms the state-of-the-art time series classification techniques in performance and efficiency, as well as notable robustness under different experiment conditions.

The proposed unknown detection scheme enhanced the feature representation of the samples, and is proved be more effective than the traditional open set classification methods such as OpenMax in distinguishing the unknown samples from the labeled ones in the time series classification problem. An additional advantage of our proposed method is that the classification and unknown detection are completed simultaneously, requiring no additional specific network structures or learning schemes.

The proposed method exhibits promising potential for future applications of non-contact sleep monitoring using millimeter wave radar.

## Figures and Tables

**Figure 1 bioengineering-11-00002-f001:**
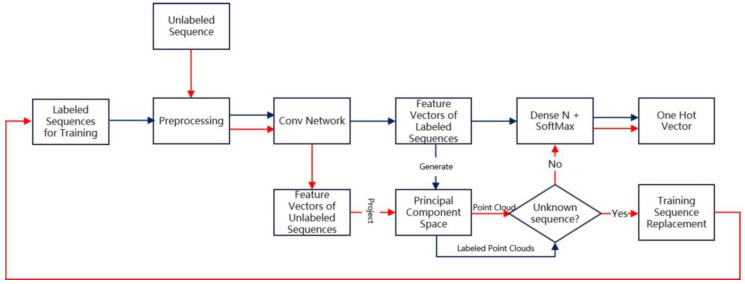
Overview of the proposed human identification scheme. The blue lines shows the training process. The red lines shows the labeling process.

**Figure 2 bioengineering-11-00002-f002:**
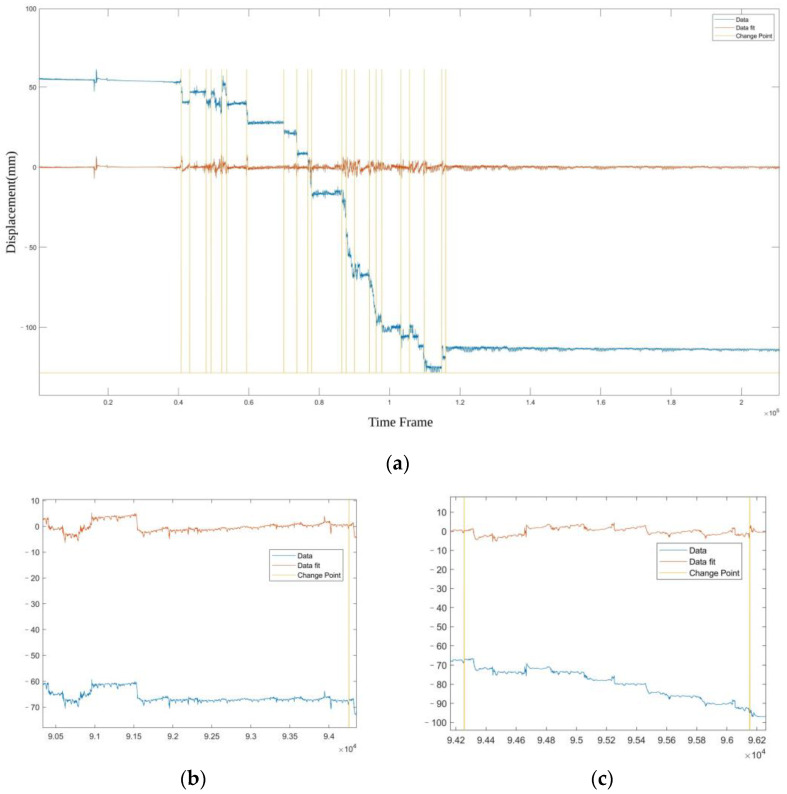
(**a**) A typical radar signal sequence. The blue line shows the original sequence. The yellow lines show the change point detected by the PELT method. The red line shows the signal after removing the linear component, it contains low amplitude and periodical component such as heat motion and respiration, which will form the sample sets for network training. (**b**) A comparison of sequences before and after baseline shift suppression. (**c**) A comparison of sequences before and after slope suppression.

**Figure 3 bioengineering-11-00002-f003:**
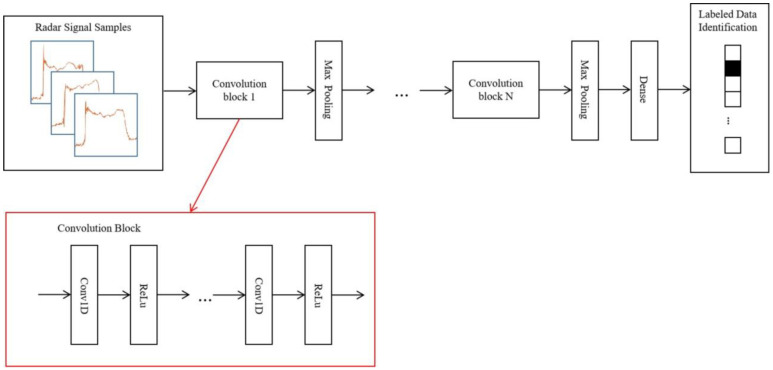
Details of the convolution neural network structure for human identification.

**Figure 4 bioengineering-11-00002-f004:**
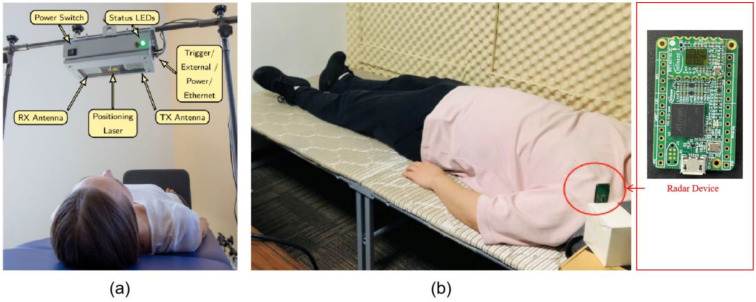
Experiment environment setup. (**a**) The data acquisition setup of the public data set. (**b**) A typical data acquisition scenario of the experimentally acquired data sets.

**Figure 5 bioengineering-11-00002-f005:**
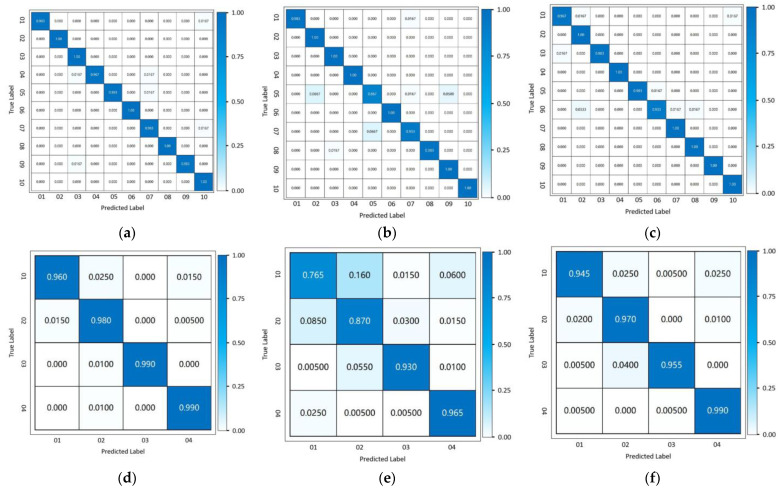
Typical confusion matrix summary. Rows: the results for public data set (**a**–**c**) and the experimentally acquired sequences (**d**–**f**). Columns: the results of our proposed method (**a**,**d**) and the comparing methods: LTSM classification network (**b**,**e**) and Inception Network (**c**,**f**).

**Figure 6 bioengineering-11-00002-f006:**
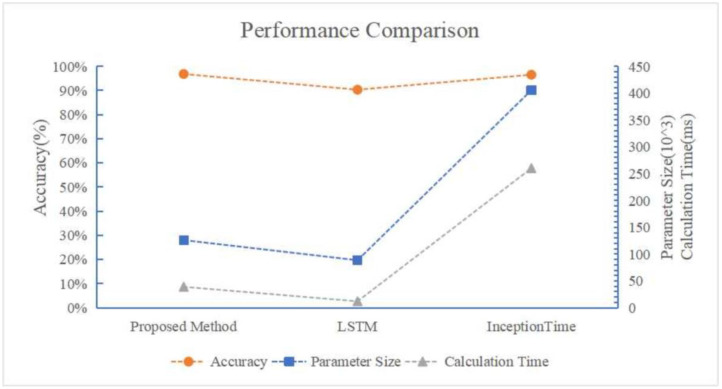
The performance comparison between the proposed network, LSTM and Inception Network.

**Figure 7 bioengineering-11-00002-f007:**
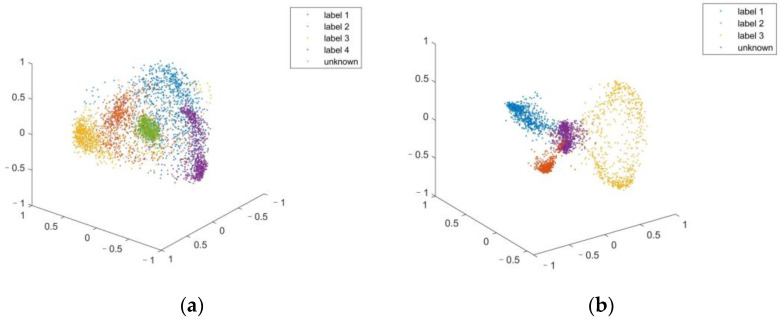
Typical PCS representation visualization. (**a**) A typical point cloud representation of labeled sequences and an unknown sequence from public data set. (**b**) A typical point cloud representation of labeled sequences and an unknown sequence from experimentally acquired data set. We can see that although only 3 principal components are used the distribution of unknown samples shows significant difference from the labeled samples.

**Figure 8 bioengineering-11-00002-f008:**
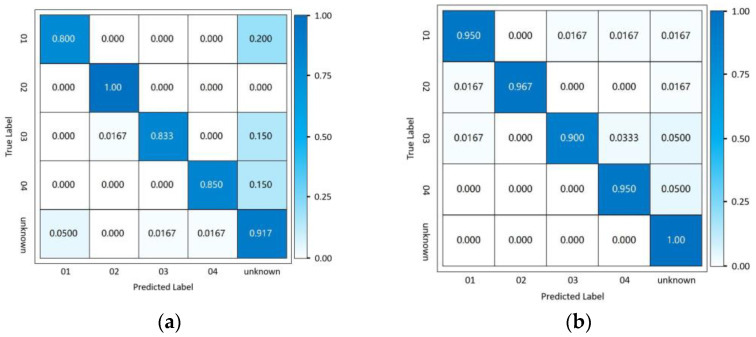
Typical confusion matrices of OpenMax (**a**) and the proposed PCS method (**b**). Compared with OpenMax, the detection accuracy of the unknown samples of the proposed method is higher, and there are less labeled samples are inaccurately detected as ‘unknown’.

**Figure 9 bioengineering-11-00002-f009:**
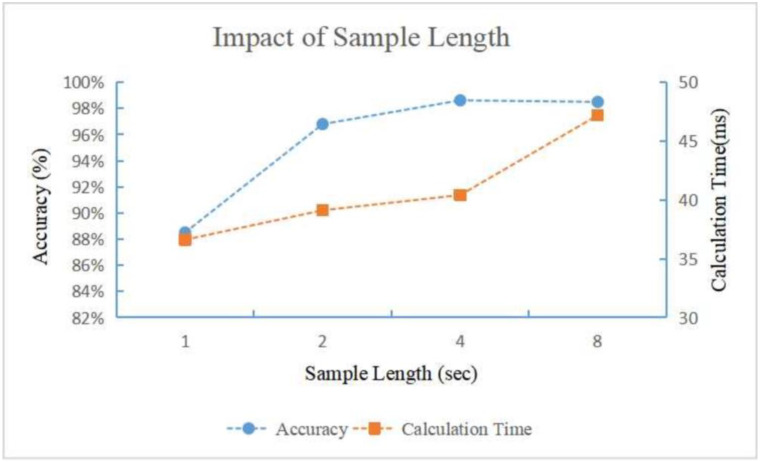
The performance-changing tendency against different segment length. We can see that to label the samples accurately the sample length should be long enough. When the sample length exceeds certain threshold, further increasing the sample length will not improve the accuracy significantly, but causing an obvious longer calculation time.

**Figure 10 bioengineering-11-00002-f010:**
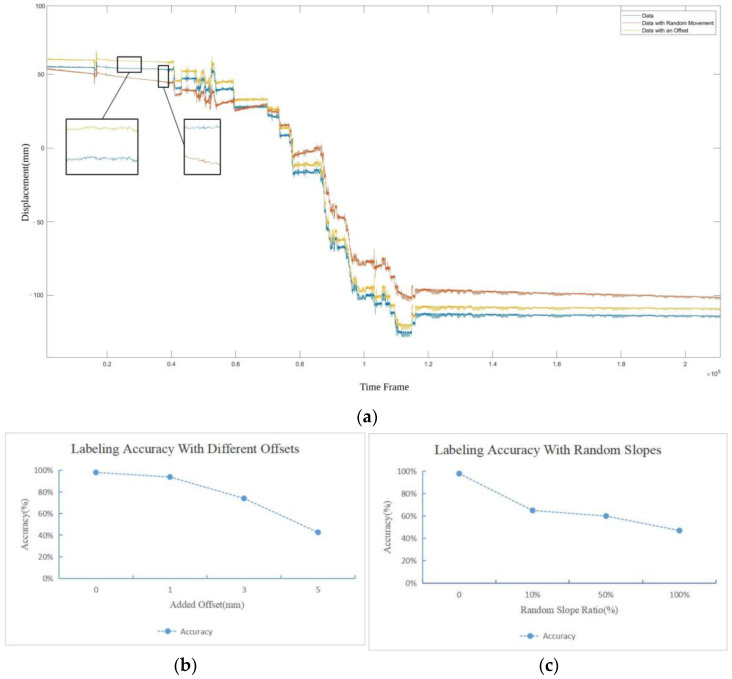
Impact of changes of data acquisition environment without data preprocessing. (**a**) A typical example of the test data set: the original sequence (blue), the sequence with offset (yellow), and the sequence with random slopes added (red); (**b**) the impact of offset on the labeling accuracy; (**c**) the impact of random slopes on the labeling accuracy.

**Figure 11 bioengineering-11-00002-f011:**
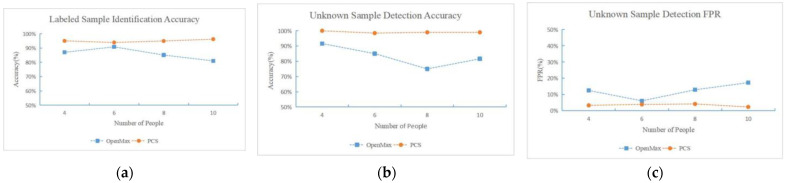
Comparison of performance against different numbers of people. (**a**) The Labeled Sample Identification Accuracy demonstrates how the open set condition affects the close set identification accuracy. (**b**) The Unknown Sample Detection Accuracy shows how many of the unknown samples can be accurately detected. (**c**) The Unknown Sample Detection FPR shows how many labeled samples are inaccurately detected as unknown.

**Table 1 bioengineering-11-00002-t001:** Summary of the radar working parameters.

Parameter	Value
Bandwidth	5.0 GHZ
Start freq	58 GHz
Chirp duration	133 us
Chirp repetition time	463 us
No. samples per chirp	128
Frame rate	100 fps
ADC sampling rate	1 MHz
Range resolution	3 cm
Velocity resolution	1.34 m/s

**Table 2 bioengineering-11-00002-t002:** Unknown data detection accuracy summary.

Test Run No.	Accuracy	Precision	Recall	FPR	F1 Score
Test 1	98.4%	97.6%	100%	5.16%	0.988
Test 2	97.5%	95.3%	99.1%	3.42%	0.961

**Table 3 bioengineering-11-00002-t003:** The impact of number of people on the identification accuracy.

No. of People	Accuracy	Total Number of Test Samples/Person	Average Number of Identified Samples/Person
4	0.983	60	59.0
6	0.969	60	58.1
8	0.989	60	59.3
10	0.987	60	59.2

## Data Availability

Data are available upon request.

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
