# Peer review of "A Deep Learning Method of Human Identification from Radar Signal for Daily Sleep Health Monitoring"

_bioengineering, 2023, doi:10.3390/bioengineering11010002_

Round 1

Reviewer 1 Report

Comments and Suggestions for Authors

The Authors have presented an approach for Daily Sleep Health Monitoring using Radar Signals. The work is very much of relevance in the medical field for sleep monitoring. Following are a few observations,

1. The impact of the number of people on the identification accuracy of human subjects is not follow any pattern, with the increased number of persons, the accuracy is fluctuating. the Author may describe the reason behind such fluctuation.

2. Figures as presented in Fig 10. needs to be improved. At present it is not readable. 

3. The Authors may explain the major reason behind the improvement in detection accuracy using their approach over the  OpenMax case. 

4. Though the work is based on radar Signal processing, the Authors have used a positioning LASER for the experiment. Authors may explain the reason behind the Laser application. The radar Sensors are being used for human detection and for sleep monitoring. 

Comments on the Quality of English Language

Acceptable. 

Reviewer 2 Report

Comments and Suggestions for Authors

Thanks for sharing an interesting study. Here are my comments.

This research proposed a daily sleep health monitoring oriented open set deep learning human identification approach using radar signal, here are some comments.

1.      The papers cited in the introduction section are not update and not related to sleep direction. Paper with transformer and sleep posture should be included. The paper below must be included.

Lai, D.K.-H.; Yu, Z.-H.; Leung, T.Y.-N.; Lim, H.-J.; Tam, A.Y.-C.; So, B.P.-H.; Mao, Y.-J.; Cheung, D.S.K.; Wong, D.W.-C.; Cheung, J.C.-W. Vision Transformers (ViT) for Blanket-Penetrating Sleep Posture Recognition Using a Triple Ultra-Wideband (UWB) Radar System. Sensors 2023, 23, 2475.

2.      The whole introduction does not address any sleep health issue or disorder and also does not provide a background of important sleep health.

3.      Figure 2 should be provided with units.

4.      Section 2.1 “The interference from the data acquisition environment is suppressed”. How is the interference suppressed? What measures had been taken to handle the clutter signal? A sample figure showing the different before and after noise suppression should be provided.

5.      What was the length of the samples chosen?

6.      The kernel size, number of filters, and number of strides in the convolution block should be provided, in addition to the number of convolution blocks.

7.      If the output of the unknown signal detection or known signal detection, it should be indicated in Figure 3.

8.      What kind of sleeping posture(s) was used in your experiment?

9.      The total sample data and distribution for training, validation and testing should be provided. Are all subjects’ data involved in training? Or, subject(s) data was held out for validation and testing?

10.   Please explain the selections of position and orientation of your radar device and how this will affect the performance of the deep learning model.

11.   The numbers in Figure 5 should align with the same significant figure. A heat map should be made to improve readability.

12.   Figure 6 is not present in professional publication standards. It must be redrawn.

13.   Section 3.3, equation numbers are missing.

14. Figures 7 to 11 are the same issues as Figures 5 and 6.

15.   Table 3 has no table heading.

16.   Novelty has not been highlight in discussion.

Comments on the Quality of English Language

This manuscript requires extensive professional English editing.

Round 2

Reviewer 2 Report

Comments and Suggestions for Authors

Thanks for the update. All questions have been addressed.